# Association of vaginal IL-4, IL-6, IL-8, IL-17, IFN-γ, and dietary intake with IBD status and vaginal microbiota in pregnant individuals

Daniela Vargas-Robles[1], Yan Rou Yap[1], Biplab Singha[1], Joyce Tien[2],
Mallika Purandare[2], Mayra Rojas-Correa[1], Camilla Madziar[1], Mellissa Picker[3],
Tina Dumont[4], Heidi K. Leftwich[4], Christine F. Frisard[5], Doyle V. Ward[1], Inga Peter[3],
Barbara Olendzki[5], Ana Maldonado-Contreras[1]*

1 Department of Microbiology, Program of Microbiome Dynamics, University of Massachusetts Chan Medical School, Worcester, Massachusetts, United States of America, 2 School of Medicine, University of Massachusetts Chan Medical School, Worcester, Massachusetts, United States of America, 3 Department of Genetics and Genomic Sciences, Icahn School of Medicine at Mount Sinai, New York, New York, United States of America, 4 Department of Obstetrics and Gynecology, Division of Maternal-Fetal Medicine, University of Massachusetts Chan Medical School, Worcester, Massachusetts, United States of America, 5 Department of Population and Quantitative Health Science, University of Massachusetts Chan Medical School, Worcester, Massachusetts, United States of America

* ana.maldonado@umassmed.edu

## Abstract

### Background

Pregnant individuals with inflammatory bowel diseases (IBD) exhibit gut inflammation and dysbiosis; however, there is limited knowledge about their vaginal environment. This is important as vaginal inflammation and high vaginal microbiota diversity are associated with adverse pregnancy outcomes.

### Objectives

We aimed to compare vaginal inflammatory markers and microbiota diversity of pregnant individuals with and without IBD in their third trimester of pregnancy and determine the role of diet in the vaginal microbiota diversity.

### Methods

We recruited pregnant individuals who provided vaginal swabs at 27–29 weeks of pregnancy. We characterized the vaginal microbiota by sequencing the V3-V4 region of the *16S rRNA* and surveyed nine key pro and anti-inflammatory cytokines by qRT-PCR from the vaginal mucosa. Participants completed three validated interviewer-led nutrition assessments of 24-hour dietary intake around the same time as the collection of vaginal samples. The nutritional assessments were used to estimate dietary quality using the validated Healthy Eating Index (HEI-2015).

**Data availability statement:** The raw sequences and associated metadata from this study have been deposited in the NCBI BioProject under accession PRJNA915128. The data are publicly available at https://www.ncbi.nlm.nih.gov/bioproject/PRJNA915128.

**Funding:** We obtained financial support from The Leona M. and Harry B. Helmsley Charitable Trust to complete the study described in the manuscript. The funders had no role in study design, data collection and analysis, decision to publish, or preparation of the manuscript.

**Competing interests:** The authors have declared that no competing interests exist.

## Results

The cohort included 23 pregnant individuals with IBD (18 with Crohn's disease and 5 with ulcerative colitis) and 25 healthy controls (HC); 56.5% of the IBD cases were in remission. Vaginal microbiota diversity and composition did not differ significantly between individuals with IBD and HC. However, the vaginal mucosa of the IBD individuals showed increased expression of Th17 pro-inflammatory cytokines (i.e., IL-6, IL-8, IL-17) and decreased expression of Th1 (IFN-γ) and Th2 (IL-4) compared to HC. Expression of IL-6 and TNF-α correlated positively with vaginal microbial diversity. The beneficial *Lactobacillus crispatus* dominated the vaginal microbiota of individuals with either high dietary quality or those consuming more vegetables or low added sugar, regardless of IBD status. In IBD cases, consumption of vegetables and added sugars were associated with reduced expression of the pro-inflammatory IFN-γ and an increased expression of anti-inflammatory IL-4.

## Conclusion

The vaginal microbiome did not differ between individuals with IBD and HC; however, IBD cases exhibit a pro-inflammatory tone in the vagina (high IL-6) that is associated with higher vaginal microbial diversity. Regardless of IBD status, healthier diets are positively associated with an increased abundance of the beneficial *L. crispatus* in the vagina.

## Introduction

Healthy pregnancies are characterized by a vaginal microbiota with low bacterial diversity dominated by *Lactobacillus* species [1]. Conversely, pregnancies with adverse outcomes are accompanied by bacterial vaginosis and a dysbiotic vaginal microbiota [2–6]. Pregnant individuals with inflammatory bowel diseases (IBD) are more likely to have bacterial vaginosis (BV) compared to healthy pregnant individuals [7]. BV is consistently associated with an increased risk of adverse perinatal outcomes, including preterm birth and low birth weight [6,8]. Also, IBD pregnant patients are themselves at higher risk of poor pregnancy outcomes [8–10]. Despite these overlapping risks, the vaginal microbiota in IBD pregnant patients has been scarcely described nor compared to healthy counterparts [7,11].

A unifying element between BV and IBD in pregnancy is the perturbation of the maternal immune balance characterized by amplification of pro-inflammatory signaling [2–5]. Disruption of the immune balance is linked to an increased risk of preterm birth [12,13]. Pregnant individuals with IBD exhibit higher levels of pro-inflammatory cytokines in serum (i.e., IL-6, IL-22, and IL-21) than healthy pregnant individuals [14]. However, there are no reports of the local vaginal immune tone of pregnant individuals with IBD.

Here, we characterized and compared the vaginal microbiota and the vaginal cytokine profiles of pregnant individuals with and without IBD. Moreover, we sought

to determine the role of environmental factors, such as diet on the vaginal microbiota composition. To our knowledge, there have been only a few studies evaluating the influence of diet on the vaginal microbiota of pregnant individuals using high throughput microbiota sequencing [15–17], yet none of the studies included all the relevant dietary components from validated instruments aiming at measuring dietary quality.

## Materials and methods

### Recruitment

We conducted a case-control study nested into our ongoing MELODY (Modulating Early Life Microbiome through Dietary Intervention in Pregnancy) trial [18]. The MELODY trial tests whether the IBD-AID™ dietary intervention during the last trimester of pregnancy can beneficially shift the microbiome of pregnant patients and their babies. The inclusion criteria included: adult pregnant individuals (18 + years old), willing to participate in the trial, being 27–29 weeks of gestation at the time of consent, with and without IBD diagnosis, and carrying a singleton pregnancy. We excluded pregnant individuals with scheduled C-section or induction of vaginal delivery before week 37 at the time of enrollment, those with medical conditions that required a special diet, HIV/AIDS+, or those unable to speak or understand English. Before the study began, we obtained study approval from the Institutional Review Board (IRB) at the University of Massachusetts Medical School (IRB protocol # H00016462) [19]. Then, we recruited women with and without IBD, at 27–29 weeks of gestation, nationwide from April 2019 through October 2020. All the women included in this nested study gave written consent and the data analyzed was obtained before any dietary intervention [18]. IBD disease activity was evaluated using validated scoring systems: the Harvey Bradshaw index [20] for participants with CD and the 6-point Mayo score [21] for participants with UC.

### Biosample collection

Vaginal and stool samples were self-collected using the OMNIgeneVAGINAL collection tube (DNA Genotek, Canada) and the ALPCO EasySampler kit (ALPCO, USA), respectively, following manufacturer instructions. Samples were kept cold until received in the lab and then frozen ~ 30h after sample collection.

### Nucleic acid isolation

DNA and RNA from vaginal samples was isolated with Dneasy PowerSoil Pro kits (QIAGEN, Germany), and PowerMicrobiome kit (QIAGEN, Germany); respectively, following the manufacturer's protocol.

### Vaginal microbiota sequencing and profiling

We performed *16S rRNA* sequencing of the V3-V4 hypervariable region as previously described [22]. Sequencing libraries were sequenced on 600 cycles using the MiSeq platform (Illumina, CA, USA). QIIME2 was used to process paired-end sequences. The DADA2 [23] algorithm, also in the QIIME2 platform, was used for quality control and obtaining representative sequences (Amplicon Sequence Variant or ASV). We used a custom database that include GreenGenes and NCBI data [24,25] for taxonomy classification. Only taxa with at least 0.1% abundance were used for the analyses, as previously done [26,27]. Also, sequences were rarefied at 7,000 sequences/sample, representing the highest number of sequences that included all vaginal samples, while maintaining a Good's coverage index > 90%. This index reflects the proportion of observed features relative to the estimated total features, ensuring sufficient sampling depth and reliable diversity estimates across all samples. S1 Table describes the sequence counts included in the analyses.

### Cytokine expression

RNA from vaginal samples was reverse transcribed using iScript cDNA Synthesis Kit (Bio-Rad, USA), and qRT–PCRs were performed using iTaq Universal SYBR Green-Supermix (Bio-Rad, USA) in an Applied Biosystems ViiA7 Real-Time PCR

machine (Thermo-Scientific, USA). Expression of each cytokine was measured in triplicate and the mean was normalized by the expression of a housekeeping gene (GAPDH). The $2^{(-\Delta\Delta Ct)}$ method was used to analyze the relative changes in gene expression. Oligonucleotides (Integrated DNA Technology, USA) used to estimate cytokine expression are listed in the S2 Table.

### Fecal calprotectin

Quantification was performed using the CalproLab ELISA ALP (Svar Life Sciences, Norway) according to the manufacturer's instructions. Total protein was quantified using the Pierce BCA Protein Assay kit (Thermo Fisher Scientific, USA). Calprotectin was normalized to initial stool weight (ng calprotectin/mg stool).

### Dietary assessment

We conducted three validated interviewer-led 24-hour dietary recalls (24HDRs) around the same time as vaginal/stool sample collection. 24HDRs were performed using the USDA Automated Multiple Pass Method [28] in conjunction with the University of Minnesota Nutrition Data for Research (NDSR) software (Version: NDS-R 2022) [29–32] as we have previously done in other studies [33–35].

### Diet quality analyses

We estimated the Healthy Eating Index 2015 (HEI-2015) from the average of the three 24HDRs as described previously by us [19]. HEI-2015 is a measure of diet quality used to assess alignment of 13 food groups with key recommendations and dietary patterns published in the *Dietary Guidelines for American*s, 2015−2020 (*Dietary Guidelines*). The population ratio method [36] was used to compute the individual scores of each of the 13 HEI-2015 components: six components with values from 0 to 5 (i.e., Total fruits, Whole fruits, Total vegetables, Greens and beans, Dairy, Total protein foods) and seven components with values from 0 to 10 (i.e., Seafood and Plant proteins, Fatty acids, Refined grains, Whole grains, Sodium, Added sugars and Saturated fats) for a total maximum score of 100 [37]. Higher values for each individual component as well as the overall HEI-2015 score represent better compliance with key recommendations in the Dietary Guidelines. Optimal intake of each of the 13 HEI-2015 dietary components are described elsewhere [37].

### Vaginal microbiota diversity analyses

Analyses were done in R using the Phyloseq package [38]. The models evaluated total or individual dietary component scores (HEI-2015) (i.e., Whole fruits, Total vegetables), or inflammatory markers (i.e., levels of fecal calprotectin or expression of vaginal cytokines), including age and body mass index (BMI), as confounder variables. Additional analyses were conducted excluding individuals using IBD medication.

Microbial alpha diversity was estimated using the Shannon and Simpson's (1-D) Indexes with the rarefied samples at the ASV level. Shannon and Simpson's Index were log-transformed to reach 'normality' of the residuals when necessary. To determine associations in alpha diversity we used linear regression models and utilized the "step" function in R to identify the best-fitted model, systematically removing non-significant variables. Alpha diversity model can be found in S1 File.

For beta diversity analyses, we employed Permutational Multivariate Analysis of Variance (PERMANOVA [39]) with adonis2 function to evaluate beta diversity measured by Aitchison distance with the non-rarefied samples at the ASV level. We initially included the variables of interest and then refined the model to include only the five variables that accounted for the most variance. Beta diversity model can be found in S2 File.

Individuals lacking BMI information (IBD = 2), fecal calprotectin measurements (IBD = 2, HC = 3) or diet (HC = 1) were excluded from the specific analysis that included those variables. Whole fruit, Fatty acid, and Seafood/plant protein were significantly collinear with Total fruit, Saturated fats, and Total vegetables, respectively (Spearman correlation coefficient $|\rho| > 0.5$). In such cases, the latter variables were retained for analyses instead of the collinear counterparts. For all analyses the threshold for significance was set at $\alpha < 0.050$.

                                                                  

### Discriminant taxa analysis

Microbial taxa (at the species level) and their association with clinical variables were assessed using MaAsLin2's [40]. Assessment was conducted only for the variables that were found to be significant or marginally significant in the beta diversity analyses and included only the taxa that was present in at least 20% of the samples.

### Community State Types (CST)

Each vaginal microbiota sample was classified into CST, as described before [41]. To compare CST by discrete variables (health status), we used Fisher exact test [42] and pairwise Fisher exact test; and by continuous variables (i.e., fecal calprotectin, cytokine expression, or dietary scores) we used Kruskal-Wallis test followed by pairwise Wilcoxon tests or ANOVA and pairwise t-test for normally or not normally distributed data, respectively; P values from pairwise analyses were adjusted for multiple comparison with the method "false discovery rate".

## Results

### The vaginal microbiota and vaginal cytokine profile of pregnant individuals by IBD status

A total of 48 pregnant individuals in their third trimester were enrolled in the study: 23 with diagnosed IBD (n = 18 Crohn's disease or CD, and n = 5, ulcerative colitis or UC) and 25 healthy controls (HC). Participants' demographics and clinical information are summarized in Table 1. Briefly, participants' mean age was 33.8 years, most had normal BMI (41.7%) or were overweight (37.5%), and most self-identified as White (93.8%). Only a few participants reported gestational diabetes or the use of antibiotics during pregnancy. None of the demographics and clinical characteristics differed by health status (IBD vs. HC, Table 1) or IBD diagnosis (CD vs. UC, S3 Table). Although more than 50% of the IBD participants were in remission at the time of recruitment, IBD participants exhibited higher fecal calprotectin levels than HC (P < 0.050, Fig 1A), as seen in a previous study [43]. Fecal calprotectin is a robust marker of gut inflammation [44].

We then sought to characterize the expression of Th1, Th2, and Th17 cytokines in the vaginal mucosa, as some of those cytokines have been associated with poor pregnancy outcomes [2,5,45]. We observed significant differences in cytokine expression by health status. Compared to HC, pregnant individuals with IBD exhibited higher expression of Th17 pro-inflammatory cytokines, specifically IL-6, IL-8, and IL-17. Conversely, IBD individuals exhibit lower expression of Th1 and Th2 cytokines: IFN-γ and IL-4, respectively (P < 0.050, Fig 1B). These results remained consistent after excluding patients receiving IBD medication (S2 Fig). We classified the vaginal microbiota of the individuals in the study into Community State Types or CST. The majority of pregnant individuals exhibited CST-I (*L. crispatus*-dominated, n = 21), followed by CST-III (*L. iners*-dominated, n = 14), CST-II (*L. gasseri*-dominated, n = 7), CST-IV (non-*Lactobacillus*-dominated, n = 4), and CST-V (*L. jensenii*-dominated n = 2) [46]. There were no differences in CSTs by health status, cytokine expression, or fecal calprotectin levels (P > 0.050, S4 Table).

Expression of IL-6 was positively associated with vaginal microbial diversity as measured by the Simpson index, but not by the Shannon index (Fig 2). In contrast, TNF-α expression was significantly associated with vaginal microbial diversity according to both the Shannon and Simpson indices, regardless of IBD status (P < 0.050, Fig 2). There were no differences in vaginal microbial diversity (alpha diversity) or composition (beta diversity) between pregnant individuals by health status (Fig 1C–1E) or by fecal calprotectin levels (P > 0.050; S1 Fig). These results remained consistent after excluding patients receiving IBD medication (P > 0.050; S2 Fig).

In summary, the vaginal microbiota of pregnant individuals with IBD (with over a half in remission but with higher fecal calprotectin levels than controls) does not differ from their healthy control counterparts, yet, the expression of important pro-inflammatory cytokines, namely IL-6, IL-8, and IL-17, is increased while expression of IFN-γ and IL-4 is decreased in the vaginal mucosa of pregnant individuals with IBD. Higher TNF-α and IL-6 expression in the vaginal mucosa is positively associated with higher vaginal microbial diversity.

**Table 1. Demographic and clinical characteristics of pregnant individuals with and without inflammatory bowel disease (IBD) included in the study and recruited between 2019 and 2022.**

| Demographic and clinical characteristics | Pregnant individuals with IBD (N = 23) | Pregnant individuals without IBD – healthy controls (N = 25) | Total (N = 48) | P value* |
|---|---|---|---|---|
| **Age** | | | | *0.414* |
| Mean (SD) | 33.3 (4.63) | 34.4 (4.97) | 33.8 (4.79) | |
| Median [Min, Max] | 33.0 [22.0, 41.0] | 36.0 [22.0, 42.0] | 34.0 [22.0, 42.0] | |
| **BMI categories \*\*** | | | | *0.072* |
| Underweight | 1 (4.3%) | 0 (0%) | 1 (2.1%) | |
| Normal | 12 (52.2%) | 8 (32.0%) | 20 (41.7%) | |
| Overweight | 8 (34.8%) | 9 (36.0%) | 17 (35.4%) | |
| Obese | 1 (4.3%) | 7 (28.0%) | 8 (16.7%) | |
| Information unavailable | 1 (4.3%) | 1 (4.0%) | 2 (4.2%) | |
| **Race** | | | | *0.490* |
| White | 23 (100%) | 22 (88.0%) | 45 (93.8%) | |
| Asian | 0 (0.0%) | 1 (4.0%) | 1 (2.1%) | |
| Other | 0 (0.0%) | 2 (8.0%) | 2 (4.2%) | |
| **Gestational diabetes** | | | | *0.098* |
| Yes | 3 (13.0%) | 0 (0%) | 3 (6.3%) | |
| No | 16 (69.6%) | 21 (84.0%) | 37 (77.1%) | |
| Information unavailable | 4 (17.4%) | 4 (16.0%) | 8 (16.7%) | |
| **Use of antibiotic during all pregnancy** | | | | *0.468* |
| No | 19 (82.6%) | 22 (88.0%) | 41 (85.4%) | |
| Yes | 3 (13.0%) | 3 (12.0%) | 6 (12.5%) | |
| Information unavailable | 1 (4.3%) | 0 (0%) | 1 (2.1%) | |
| **Preterm** | | | | *0.106* |
| No | 15 (65.2%) | 17 (68.0%) | 32 (66.7%) | |
| Yes | 4 (17.4%) | 0 (0.0%) | 4 (8.3%) | |
| Information unavailable | 4 (17.4%) | 8 (32.0%) | 12 (25.0%) | |
| **Infant birth weight (g)** | | | | *0.151* |
| Mean (SD) | 3150 (470) | 3330 (662) | 3240 (579) | |
| Median [Min, Max] | 3230 [1810, 3710] | 3290 [1080, 4520] | 3230 [1080, 4520] | |
| **Clinical characteristics relevant only for pregnant individuals with IBD** | **Pregnant individuals with IBD (N = 23)** | | | |
| **IBD diagnosis** | | | | |
| Crohn's disease | 18 (78.3%) | | | |
| Ulcerative colitis | 5 (21.7%) | | | |
| **IBD disease activity \*\*\*** | | | | |
| Mild disease | 6 (26.1%) | | | |
| Remission | 13 (56.5%) | | | |
| Information unavailable | 4 (17.4%) | | | |
| **Use of IBD medication** | | | | |
| No | 9 (39.1%) | | | |
| Yes | 14 (60.9%) | | | |

\* Fisher's exact test for categorical variables and Wilcoxon test for continuous variables.

\*\* BMI, body mass index, categories correspond to the WHO's classifications: Underweight (<18.5), normal weight (18.5–24.9), overweight (≥25.0), and obese (≥30).

\*\*\* Disease activity was estimated using the Harvey Bradshaw Index and the Mayo score for individuals with CD or UC, respectively.

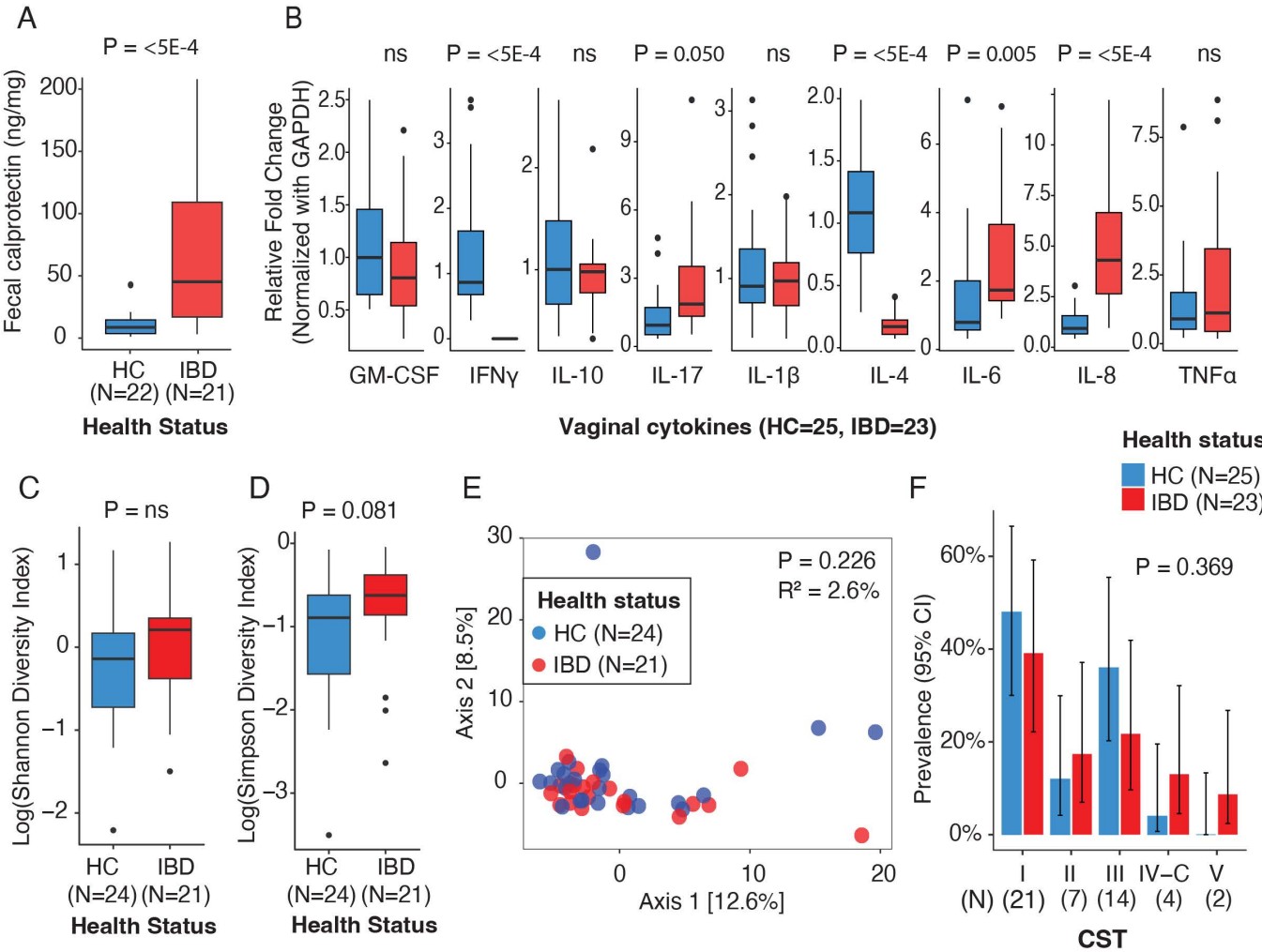

**Fig 1. Gut and vaginal inflammatory markers and vaginal microbiome diversity in pregnant individuals with and without IBD. (A)** Fecal calprotectin levels. **(B)** Vaginal cytokine gene expression. **(C,D)** Vaginal microbial alpha diversity using Shannon (C) and Simpson (D) indexes. **(E)** Principal Coordinates Analysis (PCoA) showing vaginal microbiota beta, PERMANOVA was used for beta diversity analysis based on Aitchison distance to calculate P-values and R². **(F)** Vaginal Community State Types (CSTs) proportions compared using Fisher's Exact Test. Differences in sample size (N) across panels are due to missing data for specific variables.

## Association of diet with vaginal microbiota among pregnant individuals with and without IBD

Diet is the main driver of gut microbiota diversity, and it has been suggested as an influence on vaginal microbiota composition [15–17]. Thus, we further evaluated the association of dietary quality with the vaginal microbiota.

Dietary quality in this cohort, measured by HEI-2015 score, had a mean of 63.8 out of 100, which is comparable to the average of 63.0 reported by pregnant individuals in the US [37,47]. We found no significant differences in the dietary quality or any of the individual dietary components of HEI-2015 by health status (P > 0.050, S5 Table).

Since neither dietary quality nor vaginal microbiota differ by health status, we sought to investigate associations between the microbiota and dietary quality, regardless of health status. Optimal intake of added sugar, as assessed by the HEI-2015 Added Sugar component score, was significantly associated with lower vaginal microbial diversity when measured by the Simpson index, but not by the Shannon index (P = 0.031; Fig 3A and 3B). In contrast, optimal dairy intake,

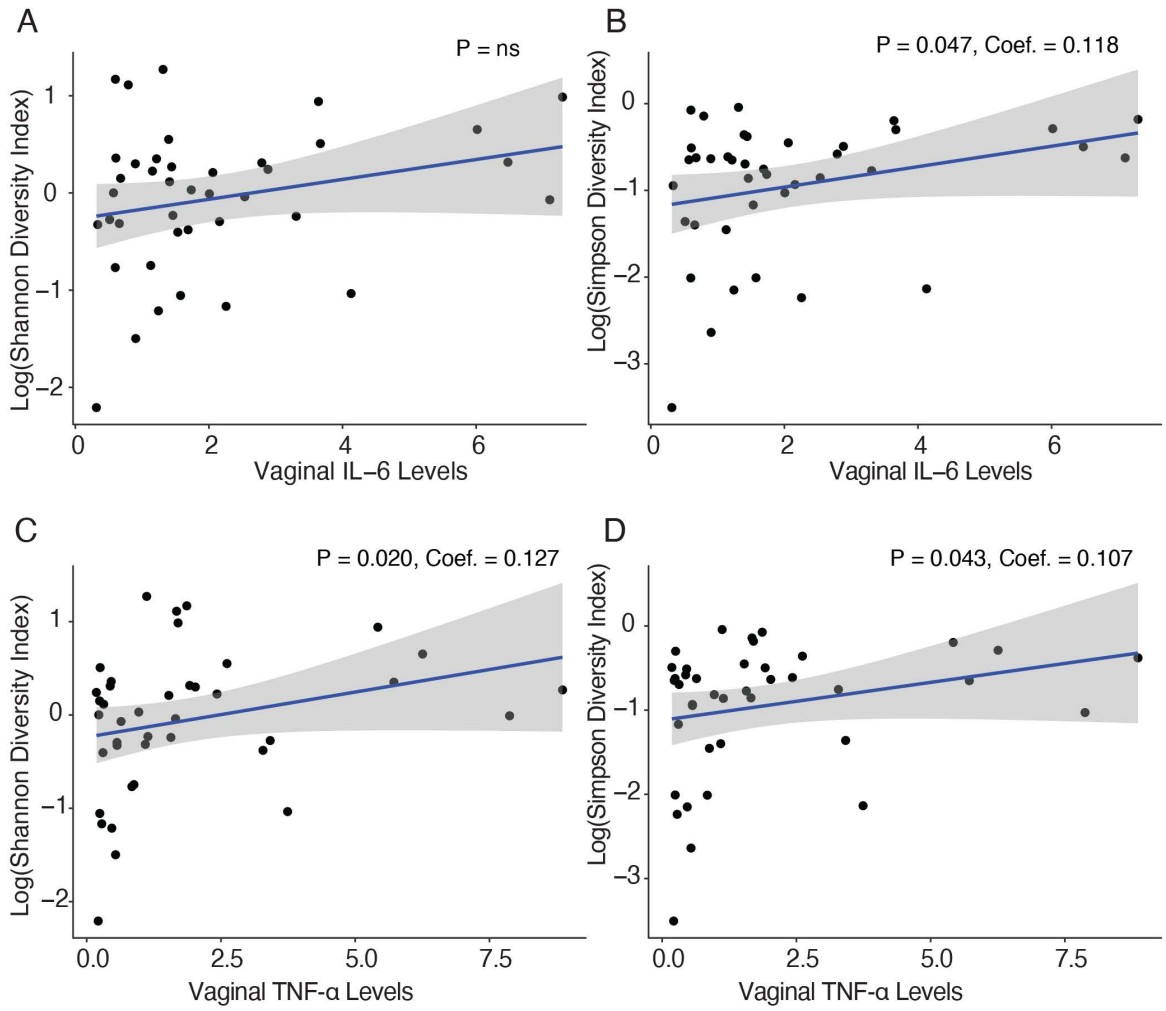

**Fig 2. Correlation between vaginal microbiota diversity and vaginal IL-6 and TNF-α expression. (A, B)** Vaginal IL-6 expression levels by Shannon (A) and Simpson (B) diversity indexes. **(C, D)** Vaginal TNF-α expression levels by Shannon (C) and Simpson (D) diversity indexes. P-values for alpha diversity were derived from linear models. The blue line indicates the fitted regression model, with the shaded area representing the 95% confidence interval.

reflected by higher HEI-2015 Dairy component scores, was significantly correlated with lower vaginal microbial diversity according to both the Simpson and Shannon indices (P < 0.050; Fig 3C and 3D).

HEI-2015 Added sugar was the only dietary component associated with vaginal microbiota composition (P = 0.002, Fig 3E). Specifically, lower consumption of added sugar (high/optimal score) was associated with a higher abundance of the beneficial *L. crispatus* and a lower abundance of *Dialister spp.*, *L. iners*, *Prevotella corporis*, *Peptoniphilus spp.*, *ph2* (family *Tissierellaceae,* best NCBI BLAST scores to *Levyella massiliensis*), and *Peptoniphilus lacrimalis* (adjusted P < 0.100, Fig 3F).

There were also significant differences across CSTs in dietary quality (P = 0.035), consumption of vegetables (P = 0.007), fruits (P = 0.016), and added sugar (P = 0.020, S6 Table). In a pairwise analysis across CST, we observed that individuals with CST-I (*L. crispatus*-dominated, N = 21) showed a higher dietary quality than individuals with a CST-III (P = 0.028, *L. iners*-dominated, N = 14. Fig 4A). Moreover, we observed that both optimal consumption of vegetables

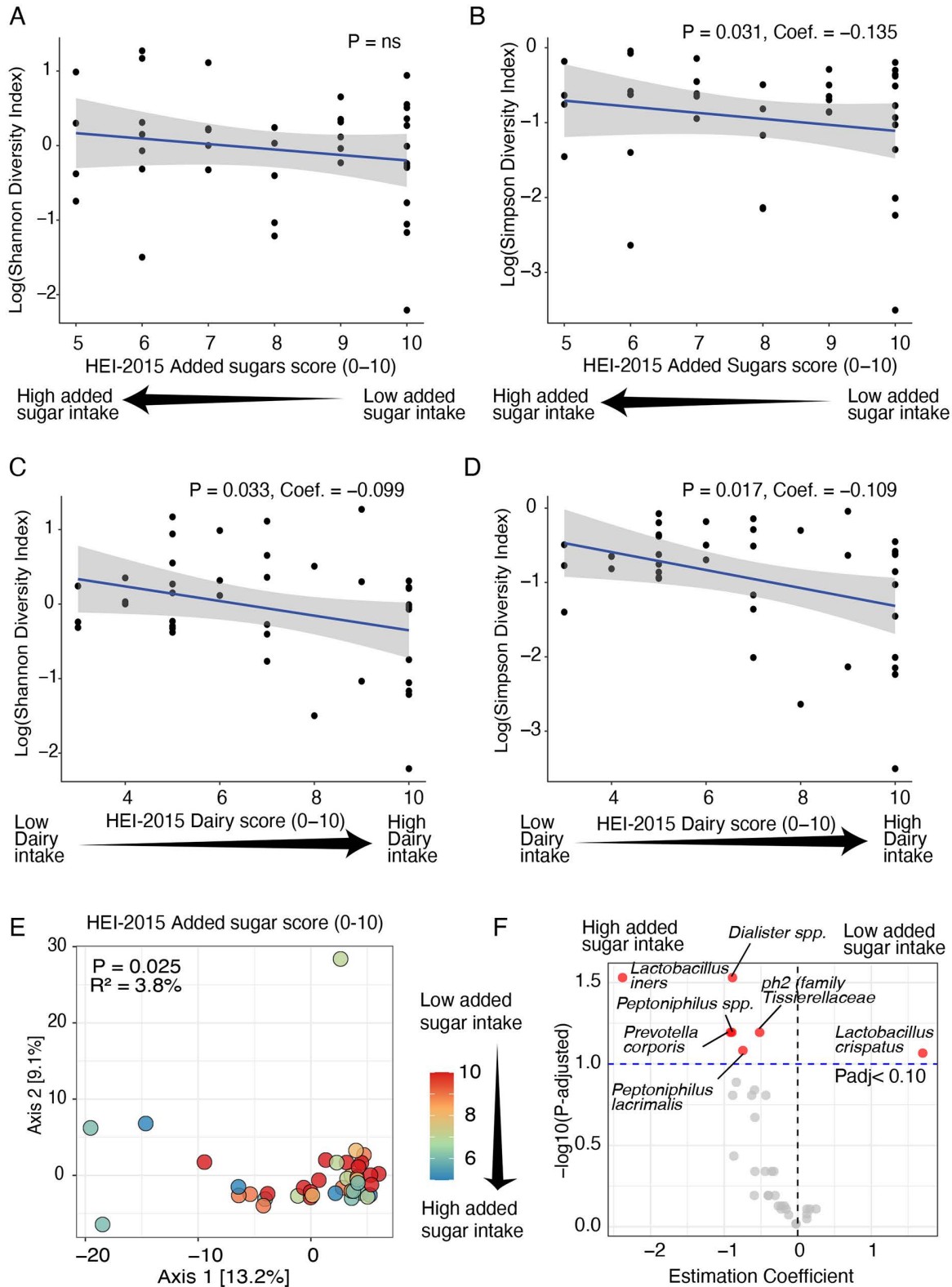

**Fig 3. Association of vaginal microbiota with dietary components. (A, B)** HEI-2015 Added sugar score associated with Shannon (A) and Simpson (B) diversity indexes. **(C, D)** HEI-2015 Dairy scores associated with Shannon (C) and Simpson (D) diversity indexes. P-values for alpha diversity were

derived from linear models. **(E)** Principal Coordinates Analysis (PCoA) showing vaginal microbiota beta diversity by HEI-2015-Added Sugar score (color gradient bar). PERMANOVA was used for beta diversity analysis based on Aitchison distance to calculate P-values and R². **(F)** Volcano plot showing significant associations between six microbial taxa and HEI-2015 Added sugar score (MaAsLin2). Significant taxa are highlighted in red based on a false discovery rate (FDR)-adjusted P-value threshold of 0.10, with taxa showing negative estimation coefficients (left) indicating negative correlations and positive coefficients (right) indicating positive correlations.

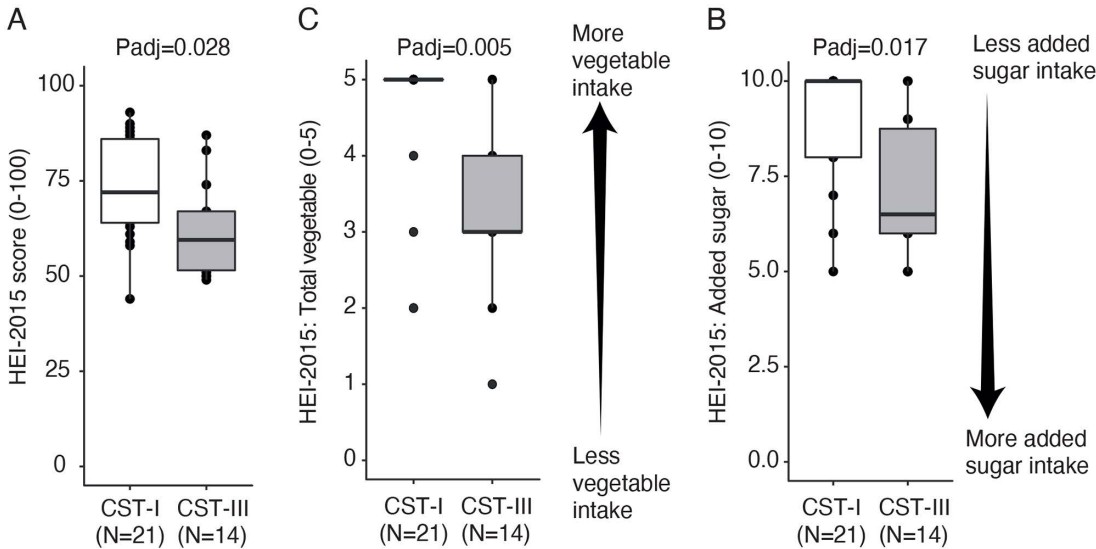

**Fig 4. Association between vaginal Community State Types (CSTs) and diet.** CST-I (*L. crispatus*-dominated) and CST-III (*L. iners*-dominated) comparison for **(A)** HEI-2015 scores (dietary quality), **(B)** HEI-2015 Total vegetables scores, and **(C)** HEI-2015 Added sugar scores. P-values correspond to post hoc pairwise comparisons between CSTs, conducted after detecting significant overall differences across all CSTs using ANOVA or Kruskal-Wallis tests (see S6 Table). Arrows indicate the direction of actual consumption for the added sugar and total vegetables to facilitate interpretation.

measured by the HEI-2015 Total vegetables component score (standard for maximum score: ≥ 1.1 cups equivalents/1,000 kcal; standard for a minimum score of zero: No vegetable consumption; P = 0.005, Fig 3B) and consumption of added sugar was significantly higher in individuals with CST-I than those with CST-III (P = 0.017, Fig 3C). However, differences in fruit consumption did not remain significant after pairwise testing across CSTs and p-value correction (adjusted P < 0.050; S6 Table).

In summary, the vaginal microbiota diversity and composition in this study can be explained by consumption of added sugars and dairy. Moreover, the dominance of the beneficial *L. crispatus* in the vaginal microbiota was observed in pregnant individuals with higher dietary quality, high vegetable intake, or low added sugar intake.

## Association of diet with vaginal cytokine expression by health status

While direct evidence linking specific dietary components to vaginal cytokine expression is limited, a couple of studies suggest that diet can indirectly influence the vaginal environment by modulating the vaginal microbiota and acting as an estrogen-agonist [48,49]. Since there were differences in vaginal cytokine expression by health status, we sought to investigate associations between the cytokine expression and diet by health status.

In the IBD cases, higher consumption of vegetables and whole grains measured by HEI-2015 Whole grains component score (standard for maximum score: ≥ 1.5 oz equiv. per 1,000 kcal; standard for a minimum score of zero: No Whole grains) were negatively associated with the expression of IFN-γ (P = 0.042) and IL-8 (P = 0.026) in the vaginal mucosa.

Conversely, higher consumption of dairy (P = 0.035) or decreased consumption of refined grain measured by HEI-2015 Refined grains component score (standard for maximum score: ≤ 1.5 oz equiv. per 1,000 kcal; standard for a minimum score of zero: ≥ 3.4 oz equiv. per 1,000 kcal; P = 0.046) were linked to higher vaginal IL-8 expression. Finally, lower added sugar intake was linked to higher vaginal IL-4 expression (P = 0.049, Fig 5).

In the healthy control group, consumption of protein measured by HEI-2015 Total protein component score (standard for maximum score: ≥ 2 oz equiv. per 1,000 kcal; standard for a minimum score of zero: No Protein Foods) was positively associated with higher expression of IFN-γ (P = 0.013) while higher consumption of refined grains was negatively associated with expression of IFN-γ (P = 0.032, Fig 5).

These findings suggest that dietary components might exert unique influences on cytokine expression in the vaginal mucosa, with specific nutrients impacting cytokine levels differently depending on the health status.

## Discussion

Our study provides novel insights into the interplay between vaginal microbiota, cytokine profiles, and diet in pregnant individuals with and without IBD. Although more than half participants with IBD were in remission, they still exhibited significantly elevated levels of gut inflammation–measured by fecal calprotectin, a robust marker of gut inflammation [44]–and a heightened pro-inflammatory immune tone in the vaginal environment. To the best of our knowledge, our study is the first to assess vaginal inflammatory markers in pregnant IBD patients.

High levels of IL-6 in intraamniotic and/or cervicovaginal fluids have been both strongly associated with spontaneous preterm labor [50–54], as well as with IBD in pregnant individuals when measured in serum [14]. Consistent with these trends, our IBD cohort exhibited elevated expression of IL-6 (and the pro-inflammatory IL-8) compared to HC, in the vaginal mucosa. This result highlights the potential application of IL-6 as a biomarker as well as a therapeutic target against poor pregnancy outcomes in the IBD population. In fact, recent work demonstrated that blockage of the IL-6 receptor abrogates preterm labor in mice [54]. In contrast, vaginal TNF-α expression—a cytokine typically elevated IBD patients—were similar between the IBD and control groups. Interestingly, overall, we observed that pregnant individuals exhibiting

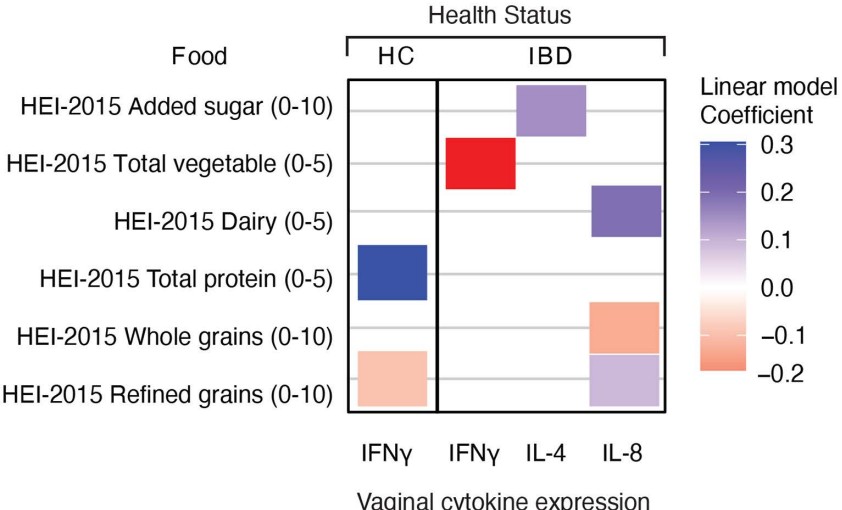

**Fig 5. Associations between HEI-2015 dietary components and vaginal cytokine expression in individuals with and without IBD.** Heatmap showing the significant correlations between HEI-2015 dietary components and the expression of three vaginal cytokines by health status (HC and IBD). Coefficients from linear models indicate the strength and direction of the associations, with positive (in blue) and negative (in red) values denoting direct and inverse correlations, respectively.

higher expression of IL-6 or TNF-α in the vaginal mucosa also had increased vaginal microbial diversity consistent with previous reports [55–58]. High vaginal microbial diversity relates to unhealthy microbial states with increased risk for BV (reviewed in [59]) and preterm birth [8,60].

No significant differences were observed in IL-1β (a potent IL-6 inducer), which are typically elevated in BV, preterm birth and associated with vaginal dysbiosis (reviewed in [51,61]). This suggests that the IL-6 elevation in IBD pregnancies may originate from alternative inflammatory pathways (i.e., NF-κβ, JAK/STAT3), or tissue sources (i.e., endometrium, decidua, trophoblasts in the placenta, fetal membranes). Our data might, at least in part, explain the increased risk of preterm birth of pregnant individuals with IBD, even when in remission or with mild disease [62,63].

Moreover, we observed significantly lower levels of both IFN-γ and IL-4 in vaginal mucosa of pregnant individuals with IBD compared to HC. Lower vaginal IFN-γ and IL-4 expression has been previously associated with increased risk of preterm birth [45]. However, also IFN-γ have been found increased on serum of IBD pregnant individuals compared to controls [64] suggesting that while systemic immune responses in IBD during pregnancy might be distinct, the local vaginal immune environment could be altered in a way that mirrors conditions associated with adverse outcomes such as preterm birth. This highlights the importance of assessing compartment-specific immune profiles to fully understand pregnancy outcomes in IBD.

Despite differences in the vaginal immune profile, the vaginal microbiota of pregnant individuals with IBD was comparable to those without the disease. This aligns with Bar et al. [65], who reported no differences in vaginal microbiota of non-pregnant individuals with and without IBD, even as disease severity progressed over time. Interestingly, Bar et al. noted a higher prevalence of vulvovaginal symptoms in IBD patients despite similar microbial communities, suggesting mucosal immune function, rather than microbiota composition, may be more affected, consistent with our findings of distinct immune profiles in IBD pregnant individuals. Contrary to our findings, Rosta et al. [7] reported that pregnant individuals with IBD more frequently exhibited abnormal vaginal microbiota compared to healthy controls. However, their methodology differed, as they used Gram-stained vaginal smears analyzed microscopically.

There is limited research on the impact of diet on vaginal health. Miller *et al.* hypothesized that high levels of starch in human diets have induced an increased amount of glycogen within the vagina, which in turn promotes the proliferation of *Lactobacillus* [66]. A vaginal microbiota dominated by *Lactobacillus* species is associated with lower levels of pro-inflammatory cytokines such as IL-1, IL-8, TNF-α, and IFN-γ [67]. Similarly, short-chain fatty acids, produced by bacterial fermentation of dietary fiber, can regulate cytokine production in the vaginal mucosa [48,49].

There is some evidence that "unhealthy diets" (i.e., high in sweets, fats, refined grains, and meat) can increase the risk of BV [68–70]. In contrast, fiber-rich diets are inversely associated with the low-Lactobacillus vaginal profile that characterizes BV [71]. Dominance of *L. crispatus* in the vagina has been observed in individuals consuming low-fat dairy, yogurt, and dietary vitamin D [17]. Similarly, high vegetable consumption and low intake of sweetened beverages have been positively associated with the dominance of *L. crispatus* in pregnant individuals [17]. In our cohort, individuals with higher dietary quality, higher vegetable intake, or lower intake of added sugars exhibited a vaginal microbiota profile dominated by the beneficial *L. crispatus*. The vaginal microbiota profile of those with lower dietary quality was dominated by *L. iners*. Compared to other *Lactobacillus* species, *L. crispatus* creates a highly acidic vaginal niche (pH < 4.5) inhospitable to BV-related bacteria [72]; thus, its dominance offers the most protective benefits against BV compared to other *Lactobacillus* species, with *L. iners* offering the least protective benefits (reviewed in [73]).

To our knowledge, no previous study has examined the relationship between vaginal cytokines and diet (and even less so in pregnant individuals with IBD). While prior research has focused on intestinal tissue and serum of IBD patients, the vaginal environment during pregnancy has remained unstudied in this context until now (reviewed in [74]. In our study, the expression of pro-inflammatory cytokines (i.e., IFN-γ and IL-8) in the vaginal mucosa of pregnant individuals with IBD, is decreased in those consuming more vegetables and whole grains (i.e., oats, brown rice, quinoa) but increased on those reporting more consumption of dairy and less consumption of refined grains (white bread, white

rice, white flour products). The expression of anti-inflammatory IL-4 was increased on individuals of this cohort reporting lower added sugar intake. Of note, several IBD-friendly diets, including the IBD-AID™ being tested in the parent MELODY study, emphasizes the consumption of vegetables and whole grains while limits the intake of dairy, refined grains, and foods with added sugars. While these foods are encourage/discourage due to their expected impact in reverting intestinal inflammation, disease symptoms, and gut dysbiosis [75–78]; our results points to the benefits they might have beyond the gut, especially for pregnant women with IBD as we did no see the same association for healthy individuals in this cohort. In healthy individuals, the expression of the pro-inflammatory IFN-γ was higher on those consuming more proteins and less refined grains.

A few key limitations constrain our study. The modest sample size for both IBD and HC cohorts curtails the statistical comparisons, particularly when several confounding variables, such as age, BMI, diet, IBD medications, are considered. While we excluded participants on active IBD medication in sensitivity analyses, residual effects of past or intermittent use may still influence results. These factors reduce the statistical power of the study. We also acknowledge that unmeasured confounders, such as lifestyle, hormonal changes, stress, or undiagnosed infections, could have influenced the findings. Additionally, the IBD samples predominantly represent individuals in remission or with mild CD and only a few participants with UC, which narrows the scope of our conclusions to this specific severity level of IBD and IBD diagnosis. Moreover, the ethnic/racial composition of our study sample, who are mainly White, introduces a limitation since the vaginal microbiota is known to vary with race and ethnicity [46]. Finally, as samples were collected only in the third trimester, we could not assess temporal changes in vaginal microbiota or cytokine expression throughout pregnancy. Our analysis focused on taxonomic profiles and did not explore the functional activity of identified microbes. Additionally, the cross-sectional design limits our ability to infer causal relationships between IBD, vaginal microbiota, and cytokine expression. These limitations suggest the need for future studies that includes pregnant individuals with IBD experiencing a range of different disease activities, and of diverse ethnic/racial backgrounds.

## Supporting information

**S1 File. Statistical models evaluating the relationship between alpha diversity indexes and explanatory variables.** Full and refined models for alpha diversity indexes (Shannon and Simpson). The full model was refined using stepwise selection. Detailed coefficients, residuals, and goodness-of-fit metrics are included.
(TXT)

**S2 File. Beta diversity models using PERMANOVA to evaluate microbial community composition.** Full and refined models for beta diversity analysis using Aitchison distance. The full model includes variables Age, BMI, calprotectin, HEI-2015 components, and vaginal cytokines, while the refined model includes the five most influential predictors. Outputs include R², F-statistics, and p-values from 999 permutations.
(TXT)

**S1 Fig. Vaginal microbiota diversity and composition in relation to fecal calprotectin levels.** (A, B) Correlation of fecal calprotectin levels with vaginal microbiota alpha diversity indexes, Shannon (A) and Simpson (B). "ns" indicates that fecal calprotectin was not retained in the final linear model. (C) Beta diversity for vaginal microbiota using PCoA based on Aitchison distances colored by fecal calprotectin levels (ng/mg).
(PDF)

**S2 Fig. Vaginal microbiota diversity, composition, and cytokine gene expression in IBD patients without IBD medication and HC.** (A, B) Comparison of vaginal microbiota alpha diversity indexes, Shannon (A) and Simpson (B), between HC and IBD patients not receiving medication. (C) Beta diversity visualization of vaginal microbiota using PCoA based on Aitchison distances, colored by health status. (D) Relative fold changes of vaginal cytokine gene expression normalized

to GAPDH in HC and IBD groups (no medication). Boxplots show median values and interquartile ranges. * $p < 0.05$, ****$p < 0.0005$.
(PDF)

**S1 Table. Overview of sequence counts and variants in 16S vaginal samples.** Details the total number of sequences and amplicon sequence variants (ASVs) in 16S vaginal samples.
(PDF)

**S2 Table. Oligonucleotide sequences used for determining vaginal cytokine gene expression.** List of primers (forward and reverse) targeting cytokine genes for qRT-PCR analysis, with sequences displayed in the 5' to 3' orientation.
(PDF)

**S3 Table. Demographic and clinical variables for pregnant individuals with Crohn's Disease (CD) or Ulcerative Colitis (UC).** Demographic and clinical characteristics of pregnant individuals with CD or UC recruited for the study between 2019 and 2022.
(PDF)

**S4 Table. Distribution of vaginal microbial Community State Types (CSTs) by health status, fecal calprotectin levels, and vaginal cytokine expression.** Distribution of CSTs across different health statuses, and mean and median or fecal calprotectin levels, and cytokine expression levels in vaginal samples. CST I is dominated by Lactobacillus crispatus, CST II by L. gasseri, CST III by L. iners, CST IV-C by a diverse set of anaerobes, and CST V by L. jensenii.
(PDF)

**S5 Table. HEI-2015 scores and dietary components by health status.** HEI-2015 and individual dietary component scores for individuals with IBD and HC.
(PDF)

**S6 Table. HEI-2015 scores and dietary components distribution across vaginal Community State Types (CSTs).** HEI-2015 scores and individual dietary component distributions among vaginal CSTs. CST I is dominated by *L. crispatus*, CST II by *L. gasseri*, CST III by *L. iners*, CST IV-C by diverse anaerobes, and CST V by *L. jensenii*.
(PDF)

## Acknowledgments

We thank all the participants for their time and effort in collecting samples and responding to surveys. We also thank Rafael Lopez Martinez for contributing to designing scripts for analysis automatization.

## Author contributions

**Conceptualization:** Inga Peter, Barbara Olendzki, Ana Maldonado-Contreras.

**Data curation:** Daniela Vargas-Robles, Christine F. Frisard.

**Formal analysis:** Daniela Vargas-Robles.

**Funding acquisition:** Inga Peter, Barbara Olendzki, Ana Maldonado-Contreras.

**Investigation:** Joyce Tien, Ana Maldonado-Contreras.

**Methodology:** Yan Rou Yap, Biplab Singha, Mallika Purandare, Mayra Rojas-Correa, Camilla Madziar, Doyle V. Ward.

**Project administration:** Mellissa Picker.

**Supervision:** Inga Peter, Barbara Olendzki, Ana Maldonado-Contreras.

**Visualization:** Daniela Vargas-Robles.

**Writing – original draft:** Daniela Vargas-Robles, Ana Maldonado-Contreras.

**Writing – review & editing:** Yan Rou Yap, Biplab Singha, Joyce Tien, Camilla Madziar, Mellissa Picker, Tina Dumont, Heidi K. Leftwich, Inga Peter, Barbara Olendzki, Ana Maldonado-Contreras.

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
