## [Decision Letter · Decision Letter 0]

28 Apr 2025

Dear Dr. Maldonado-Contreras,

Thank you for submitting your manuscript to PLOS ONE. After careful consideration, we feel that it has merit but does not fully meet PLOS ONE’s publication criteria as it currently stands. Therefore, we invite you to submit a revised version of the manuscript that addresses the points raised during the review process.

We look forward to receiving your revised manuscript.

Kind regards,

António Machado

Academic Editor

PLOS ONE

**Journal Requirements:**

1. When submitting your revision, we need you to address these additional requirements. Please ensure that your manuscript meets PLOS ONE's style requirements, including those for file naming. The PLOS ONE style templates can be found at https://journals.plos.org/plosone/s/file?id=wjVg/PLOSOne_formatting_sample_main_body.pdf and https://journals.plos.org/plosone/s/file?id=ba62/PLOSOne_formatting_sample_title_authors_affiliations.pdf 2. Thank you for stating the following financial disclosure: IP, BO, and AMC received a grant to conduct this work from The Leona M. and Harry B. Helmsley Charitable Trusthttps://helmsleytrust.org/  Please state what role the funders took in the study.  If the funders had no role, please state: "The funders had no role in study design, data collection and analysis, decision to publish, or preparation of the manuscript." If this statement is not correct you must amend it as needed. Please include this amended Role of Funder statement in your cover letter; we will change the online submission form on your behalf. 3. Thank you for stating the following in the Acknowledgments Section of your manuscript: We thank all the participants for their time and effort in collecting samples and responding to surveys. We also thank Rafael Lopez Martinez for contributing to designing scripts for analysis automatization. The Leona M. and Harry B. Helmsley Charitable Trust supported this work. We note that you have provided funding information that is not currently declared in your Funding Statement. However, funding information should not appear in the Acknowledgments section or other areas of your manuscript. We will only publish funding information present in the Funding Statement section of the online submission form. Please remove any funding-related text from the manuscript and let us know how you would like to update your Funding Statement. Currently, your Funding Statement reads as follows: IP, BO, and AMC received a grant to conduct this work from The Leona M. and Harry B. Helmsley Charitable Trust https://helmsleytrust.org/  Please include your amended statements within your cover letter; we will change the online submission form on your behalf. 4. In the online submission form, you indicated that your data will be submitted to a repository upon acceptance.  We strongly recommend all authors deposit their data before acceptance, as the process can be lengthy and hold up publication timelines. Please note that, though access restrictions are acceptable now, your entire minimal dataset will need to be made freely accessible if your manuscript is accepted for publication. This policy applies to all data except where public deposition would breach compliance with the protocol approved by your research ethics board. If you are unable to adhere to our open data policy, please kindly revise your statement to explain your reasoning and we will seek the editor's input on an exemption. 5. When completing the data availability statement of the submission form, you indicated that you will make your data available on acceptance. We strongly recommend all authors decide on a data sharing plan before acceptance, as the process can be lengthy and hold up publication timelines. Please note that, though access restrictions are acceptable now, your entire data will need to be made freely accessible if your manuscript is accepted for publication. This policy applies to all data except where public deposition would breach compliance with the protocol approved by your research ethics board. If you are unable to adhere to our open data policy, please kindly revise your statement to explain your reasoning and we will seek the editor's input on an exemption. Please be assured that, once you have provided your new statement, the assessment of your exemption will not hold up the peer review process. 6. PLOS requires an ORCID iD for the corresponding author in Editorial Manager on papers submitted after December 6th, 2016. Please ensure that you have an ORCID iD and that it is validated in Editorial Manager. To do this, go to ‘Update my Information’ (in the upper left-hand corner of the main menu), and click on the Fetch/Validate link next to the ORCID field. This will take you to the ORCID site and allow you to create a new iD or authenticate a pre-existing iD in Editorial Manager. 7. Please include your full ethics statement in the ‘Methods’ section of your manuscript file. In your statement, please include the full name of the IRB or ethics committee who approved or waived your study, as well as whether or not you obtained informed written or verbal consent. If consent was waived for your study, please include this information in your statement as well.

**Additional Editor Comments:**

Dear authors,

I am pleased to say that both reviewers enjoyed the manuscript very much and we are excited about the possibility of publishing your work. However, both reviewers recommended several improvements to the original manuscript. Please read carefully both reviewers’ reports addressing and answering all comments and suggestions.

So, I kindly invite the authors to realize a thoughtful revision of the submitted manuscript to achieve publication endorsement by the reviewers.

Thank you and best regards,

António Machado

Reviewers' comments:

Reviewer's Responses to Questions

**Comments to the Author**

1. Is the manuscript technically sound, and do the data support the conclusions?

Reviewer #1: Yes

Reviewer #2: Yes

2. Has the statistical analysis been performed appropriately and rigorously?

Reviewer #1: Yes

Reviewer #2: Yes

3. Have the authors made all data underlying the findings in their manuscript fully available?

Reviewer #1: No

Reviewer #2: Yes

4. Is the manuscript presented in an intelligible fashion and written in standard English?

Reviewer #1: Yes

Reviewer #2: Yes

**Reviewer #1: ** The manuscript entitled "Association of vaginal inflammatory markers and diet with the vaginal microbiota of

pregnant individuals with or without IBD" by Vargas-Robles et al. is written well but there are following issues.

1. The study includes only 48 pregnant individuals, with 23 having IBD and 25 as healthy controls. The small sample size limits statistical power and generalizability.

2. The majority of participants were White (93.8%), which reduces the applicability of findings to more diverse populations with different genetic backgrounds and environmental exposures.

3.The study captures data from a single time point in the third trimester, missing potential changes in vaginal microbiota and cytokine expression across pregnancy.

4. While the study accounts for IBD status and some demographic variables, other potential confounders such as diet, lifestyle, and medication use may still influence microbiota and cytokine levels.

5. Although sensitivity analyses exclude participants on IBD medication, residual effects of past or intermittent medication use might still affect results.

6. While microbial diversity and composition are analyzed, functional roles of identified taxa (e.g., metabolic activity, immune modulation) remain unexplored.

7. The cross-sectional nature of the study prevents conclusions about causal relationships between IBD, vaginal microbiota, and cytokine expression.

8. Other factors, such as hormonal changes, stress, or undiagnosed infections, may contribute to cytokine expression and microbiota composition but were not assessed.

**Reviewer #2: ** Title: Association of vaginal inflammatory markers and diet with the vaginal microbiota of pregnant individuals with or without IBD

I really enjoyed reading this article. It is well written and follows a nice sequence of experiments to help understand the research conducted for this paper.

Summary of the paper: This study used samples from MELODY trial to study the vaginal microbiota of pregnant women with or without IBD. They analysed the samples to identify the immune profile of vaginal mucosa. The microbiota diversity and CST composition was similar for IBD and healthy individuals. However, the authors identified elevated cytokines such as IL-6, IL-8 which are associated with poorer pregnancy outcomes and lower levels of IL-4 and IFN-g. Thereafter they also studied the impact of diet – especially added sugars and dairy intake on vaginal microbiota diversity and found lower added sugars and lower dairy intake was associated with lower microbiota diversity. Interestingly lower added sugar was associated with higher levels of L. crispatus which is a beneficial microbe. Finally, they also showed association between diet and inflammatory cytokines with total vegetables and whole grain being negatively associated with expression of IFN-γ and IL-8 in pregnant women with IBD.

Review comments:

Title: Recommend to add IL-6, IL-8 and IFN-g to title (e.g. association of inflammatory markers such as IL-6…..and diet with the vaginal……) to improve impact.

Introduction:

General: Overall, this section needs to be strengthened. The introduction is not supported by sufficient data and the authors need to generate reader interest in the topic by connecting bacterial vaginosis, IBD and pregnancy outcomes. Also, there are few articles available which talk about inflammatory markers in IBD and pregnancy outcomes which can be cited to strengthen the background regarding inflammatory markers in IBD and pregnancy outcomes.

Line 54 and 56: Suggest to specify the adverse pregnancy outcomes for bacterial vaginosis and IBD. There are publications which mention that these are pre-term deliveries and low-birth weight deliveries. Also, your baseline data shows the same, so the mention of types of outcomes will support your introduction.

Line 56-57: The claim needs to be re-worded ‘They 56 are also at higher risk of poor pregnancy outcomes, yet their vaginal microbiota has not been described’. According to the ref: Hill JE, Peña-Sánchez JN, Fernando C, Freitas AC, Withana Gamage N, Fowler S. Composition and Stability of the Vaginal Microbiota of Pregnant Women With Inflammatory Bowel Disease. Inflamm Bowel Dis. 2022 Jun 3;28(6):905-911 – the microbiota is studied. It might not be studied widely, however the claim as written by you becomes untrue. In addition to that, addition of above ref will support your paper as it mentions that L. crispatus is the most commonly found microbe.

Line 62-64: This sentence is confusing and I would suggest to delete this line or find an alternative ref to support the claim. This line might be referenced to discussion part of Bar O et al (ref 12): “A few studies examining changes in vaginal microbiota over time in pregnant participants with IBD have not found significant differences from what has been described in pregnant people without IBD, aside from higher prevalence of Mollicutes in one cohort”. The Bar O et al shows that there is no association between IBD severity over time with changes in vaginal microbiome between pre and post menopausal women.

Materials and Methods:

Line 75-84: Recommend to describe MELODY trial briefly – for e.g. the MELODY (Modulating Early Life Microbiome through Dietary Intervention in Pregnancy) trial tests whether the IBD-AID™ dietary intervention during the last trimester of pregnancy can beneficially shift the microbiome of ……” from the relevant publication (Peter et al). I had to visit the publication to understand the original study and its inclusion/exclusion criteria.

Line 80: “Written consent”. Word consent is missing.

Line 127-129: Data imputation - use of median values- for missing data is not recommended. Ideally data from missing samples should be excluded or the last available data for the patient should be used. Recommend to do the analysis with excluding the patient values.

Line 131: spelling error “later”

Results:

Line 202 to 206: based on the title of the paper, I was expecting this to be first results which would be presented visually. I recommend to present this in tabular form. In addition to that, table S4 should be presented after figure 1B. Even though the results are not significant, it is important to demonstrate visually that the CST based microbiota diversity. In the table you can put a column saying P values NS. This would really strengthen the paper in terms of visually demonstrating that the authors have evaluated the data by subgroups and arrived at the conclusion which is presented in the paper.

After line 228: IL-6 results and its implication in disease is covered. However, equally interesting to note that IL-4 and IFN-g were lower. Higher IFN-g is associated with poorer pregnancy outcomes, so lower IFN-g levels in this study are interesting as the baseline characteristics show that there were 4 preterm births in IBD cohort. Exploring this would provide interesting insights.

Line 282: Do you mean fig 4? Table S4 is not showing diet associated data. Please verify.

Table S6: This data was very interesting to me. However, it is not described in the results section. HEI, total vegetable, total fruit and added sugar – all show significant association of variable to CST type I especially L. crispatus which is also mentioned in the abstract. These results should be described in the paper in a short paragraph.

General editorial: Lactobacillus is expanded at few instances and is shortened to L at few instances. Keeping it consistent would improve the editorial quality. (e.g., line 272 and 279)

Discussion:

Line 324 and 325: The authros demonstrate high fecal calprotectin (fig 1A). This well studied indicator of gut inflammation in IBD, so this needs to be mentioned in the sentence as it will improve your argument of high gut inflammation and corelate it to immune tone of vaginal environment.

Line 331-333: The role of IFN-g is not very well explored in this paper even though it was found to be lower in the study and showed association with diet. The authors state that lower IFN-g is associated with poor pregnancy outcomes, but I came across articles which stated that high IFN-g is associated with poorer outcomes. (ref: Jijon H, Ueno A, Sharifi N, Leung Y, Ghosh S, Seow CH. Elevated interferon-gamma levels during pregnancy are associated with adverse maternofetal outcomes in IBD. Gut. 2020 Oct;69(10):1895-1897 AND Hossein-Javaheri N, Youssef M, Jeyakumar Y, Huang V, Tandon P. The Management of Inflammatory Bowel Disease during Reproductive Years: An Updated Narrative Review. Reproductive Medicine. 2023; 4(3):180-197.) I am giving 2 examples here, however I suggest that authors should review the literature thoroughly to understand the role of IFN-g in pregnancy outcomes for IBD patients and thereafter elaborate on the IFN-g role in the discussion section. If this is a contradictory result, then it needs to be highlighted as a unique finding of this study.

General observation: The limitations of the study are well covered.

**Do you want your identity to be public for this peer review?** For information about this choice, including consent withdrawal, please see our Privacy Policy

Reviewer #1: No

Reviewer #2: **Yes: ** Amruta A Jambekar

---

## [Author Response · Author response to Decision Letter 1]

28 Aug 2025

PONE-D-25-04775

Association of vaginal inflammatory markers and diet with the vaginal microbiota of pregnant individuals with or without IBD

Journal Requirements:

R: We have revised all the documents to comply with PLOS ONE's style requirements

IP, BO, and AMC received a grant to conduct this work from The Leona M. and Harry B. Helmsley Charitable Trust https://helmsleytrust.org/

R: We have included the role of the funder in the cover letter.

We thank all the participants for their time and effort in collecting samples and responding to surveys. We also thank Rafael Lopez Martinez for contributing to designing scripts for analysis automatization. The Leona M. and Harry B. Helmsley Charitable Trust supported this work.

IP, BO, and AMC received a grant to conduct this work from The Leona M. and Harry B. Helmsley Charitable Trust” https://helmsleytrust.org/

R: We remove the financial statement from Acknowledgments and include it in the cover letter.

4. In the online submission form, you indicated that your data will be submitted to a repository upon acceptance. We strongly recommend all authors deposit their data before acceptance, as the process can be lengthy and hold up publication timelines. Please note that, though access restrictions are acceptable now, your entire minimal dataset will need to be made freely accessible if your manuscript is accepted for publication. This policy applies to all data except where public deposition would breach compliance with the protocol approved by your research ethics board. If you are unable to adhere to our open data policy, please kindly revise your statement to explain your reasoning and we will seek the editor's input on an exemption.

R: The data is now publicly available in the NCBI under BioProject ID PRJNA915128

R: Please see our response to comment #4 above.

6. PLOS requires an ORCID iD for the corresponding author in Editorial Manager on papers submitted after December 6th, 2016. Please ensure that you have an ORCID iD and that it is validated in Editorial Manager. To do this, go to ‘Update my Information’ (in the upper left-hand corner of the main menu), and click on the Fetch/Validate link next to the ORCID field. This will take you to the ORCID site and allow you to create a new iD or authenticate a pre-existing iD in Editorial Manager.

R: The corresponding author will need help linking two accounts on Editorial Manager. The corresponding author ORCID is 0000-0002-5967-9623

R: We have now included information about the IRB and consenting process in the Methods section of the manuscript. See lines 77-79.

Reviewer's Responses to Questions

Comments to the Author

1. Is the manuscript technically sound, and do the data support the conclusions?

Reviewer #1: Yes

Reviewer #2: Yes

2. Has the statistical analysis been performed appropriately and rigorously?

Reviewer #1: Yes

Reviewer #2: Yes

3. Have the authors made all data underlying the findings in their manuscript fully available?

The requires authors to make all data underlying the findings described in their manuscript fully available without restriction, with rare exception (please refer to the Data Availability Statement in the manuscript PDF file). The data should be provided as part of the manuscript or its supporting information, or deposited to a public repository. For example, in addition to summary statistics, the data points behind means, medians and variance measures should be available. If there are restrictions on publicly sharing data—e.g. participant privacy or use of data from a third party—those must be specified.

Reviewer #1: No

Reviewer #2: Yes

R: Please see our response to comment #4 above.

4. Is the manuscript presented in an intelligible fashion and written in standard English?

Reviewer #1: Yes

Reviewer #2: Yes

5. Review Comments to the Author

Reviewer #1:

The manuscript entitled "Association of vaginal inflammatory markers and diet with the vaginal microbiota of pregnant individuals with or without IBD" by Vargas-Robles et al. is written well but there are following issues.

1. The study includes only 48 pregnant individuals, with 23 having IBD and 25 as healthy controls. The small sample size limits statistical power and generalizability.

R: We agree. As stated in our discussion, we explicitly acknowledge that the modest sample size constrains statistical power and generalizability, and we emphasize the need for larger, prospective studies to further validate and expand upon our findings. See lines 413-415

2. The majority of participants were White (93.8%), which reduces the applicability of findings to more diverse populations with different genetic backgrounds and environmental exposures.

R: We agree and address this point in the discussion. See lines 421-422

3.The study captures data from a single time point in the third trimester, missing potential changes in vaginal microbiota and cytokine expression across pregnancy.

R: We agree with reviewer on the importance of understanding long-term dynamics of microbiome and immune markers. We have now included a statement acknowledging our single timepoint approach as a limitation of the study. See lines 422-424

4. While the study accounts for IBD status and some demographic variables, other potential confounders such as diet, lifestyle, and medication use may still influence microbiota and cytokine levels.

R: Our analyses included cofounding variables such as age, BMI, diet, and IBD medications (See lines 144-148). We now acknowledge that other unmeasured confounders may still influence results (See lines 417-418)

5. Although sensitivity analyses exclude participants on IBD medication, residual effects of past or intermittent medication use might still affect results.

R: We agree that residual effects of past or intermittent use cannot be entirely ruled out, and we have now added this clarification to the discussion. See lines 415-416

6. While microbial diversity and composition are analyzed, functional roles of identified taxa (e.g., metabolic activity, immune modulation) remain unexplored.

R: We now acknowledged in the discussion the focus on taxonomic variation not on metabolic function or immune modulation. See lines 424-426

7. The cross-sectional nature of the study prevents conclusions about causal relationships between IBD, vaginal microbiota, and cytokine expression.

R: We agree and emphasize in throughout the manuscript that our findings are associative.

8. Other factors, such as hormonal changes, stress, or undiagnosed infections, may contribute to cytokine expression and microbiota composition but were not assessed.

R: Please see response to comment #4 from this reviewer.

Reviewer #2:

Title: Association of vaginal inflammatory markers and diet with the vaginal microbiota of pregnant individuals with or without IBD

I really enjoyed reading this article. It is well written and follows a nice sequence of experiments to help understand the research conducted for this paper.

Summary of the paper: This study used samples from MELODY trial to study the vaginal microbiota of pregnant women with or without IBD. They analysed the samples to identify the immune profile of vaginal mucosa. The microbiota diversity and CST composition was similar for IBD and healthy individuals. However, the authors identified elevated cytokines such as IL-6, IL-8 which are associated with poorer pregnancy outcomes and lower levels of IL-4 and IFN-g. Thereafter they also studied the impact of diet – especially added sugars and dairy intake on vaginal microbiota diversity and found lower added sugars and lower dairy intake was associated with lower microbiota diversity. Interestingly lower added sugar was associated with higher levels of L. crispatus which is a beneficial microbe. Finally, they also showed association between diet and inflammatory cytokines with total vegetables and whole grain being negatively associated with expression of IFN-γ and IL-8 in pregnant women with IBD.

Review comments:

Title: Recommend to add IL-6, IL-8 and IFN-g to title (e.g. association of inflammatory markers such as IL-6…..and diet with the vaginal……) to improve impact.

R: Thank you for this suggestion. We have revised the title.

Introduction:

General: Overall, this section needs to be strengthened. The introduction is not supported by sufficient data and the authors need to generate reader interest in the topic by connecting bacterial vaginosis, IBD and pregnancy outcomes. Also, there are few articles available which talk about inflammatory markers in IBD and pregnancy outcomes which can be cited to strengthen the background regarding inflammatory markers in IBD and pregnancy outcomes.

R: We appreciate this insightful recommendation. We have revised the Introduction to include further citations concerning associations of bacterial vaginosis and IBD with adverse outcomes. See lines 55-58

Line 54 and 56: Suggest to specify the adverse pregnancy outcomes for bacterial vaginosis and IBD. There are publications which mention that these are pre-term deliveries and low-birth weight deliveries. Also, your baseline data shows the same, so the mention of types of outcomes will support your introduction.

R: We now include specifics on adverse outcomes. See lines 56-57

Line 56-57: The claim needs to be re-worded ‘They 56 are also at higher risk of poor pregnancy outcomes, yet their vaginal microbiota has not been described’. According to the ref: Hill JE, Peña-Sánchez JN, Fernando C, Freitas AC, Withana Gamage N, Fowler S. Composition and Stability of the Vaginal Microbiota of Pregnant Women With Inflammatory Bowel Disease. Inflamm Bowel Dis. 2022 Jun 3;28(6):905-911 – the microbiota is studied. It might not be studied widely, however the claim as written by you becomes untrue. In addition to that, addition of above ref will support your paper as it mentions that L. crispatus is the most commonly found microbe.

R: Thank you for pointing this out. We have rephrased the sentence to reflect that few studies have examined the vaginal microbiota in IBD during pregnancy. See line 59.

Line 62-64: This sentence is confusing and I would suggest to delete this line or find an alternative ref to support the claim. This line might be referenced to discussion part of Bar O et al (ref 12): “A few studies examining changes in vaginal microbiota over time in pregnant participants with IBD have not found significant differences from what has been described in pregnant people without IBD, aside from higher prevalence of Mollicutes in one cohort”. The Bar O et al shows that there is no association between IBD severity over time with changes in vaginal microbiome between pre and post menopausal women.

R: Thank you for the suggestion. We have deleted the aforementioned sentence from the Introduction as suggested. We have also clarified the results from Bar et al. (2023). See lines 367- and 372

Materials and Methods:

Line 75-84: Recommend to describe MELODY trial briefly – for e.g. the MELODY (Modulating Early Life Microbiome through Dietary Intervention in Pregnancy) trial tests whether the IBD-AID™ dietary intervention during the last trimester of pregnancy can beneficially shift the microbiome of ……” from the relevant publication (Peter et al). I had to visit the publication to understand the original study and its inclusion/exclusion criteria.

R: We now include a brief description of the inclusion/exclusion criteria. See lines 77-79.

Line 80: “Written consent”. Word consent is missing.

R: Corrected. See line 87

Line 127-129: Data imputation - use of median values- for missing data is not recommended. Ideally data from missing samples should be excluded or the last available data for the patient should be used. Recommend to do the analysis with excluding the patient values.

R: Thank you for the suggestion. We have re-analyzed the data excluding samples with missing values for fecal calprotectin (N=4) and BMI (N=2), which were the variables originally imputed. All the figures and results reflect the new analyses excluding those with missing values.

Line 131: spelling error “later”

R: Corrected

Results:

Line 202 to 206: based on the title of the paper, I was expecting this to be first results which would be presented visually. I recommend to present this in tabular form. In addition to that, table S4 should be presented after figure 1B. Even though the results are not significant, it is important to demonstrate visually that the CST based microbiota diversity. In the table you can put a column

---

## [Decision Letter · Decision Letter 1]

19 Sep 2025

Dear Dr. Maldonado-Contreras,

We look forward to receiving your revised manuscript.

Kind regards,

António Machado

Academic Editor

PLOS ONE

Journal Requirements:

Additional Editor Comments:

Dear authors,

I am pleased to inform you that both reviewers only requested some revisions for future publication endorsement. Please carefully answer reviewers' concerns and rectify the manuscript following their comments.

Thank you for choosing PLOS ONE journal and best regards,

António Machado

Reviewers' comments:

Reviewer's Responses to Questions

**Comments to the Author**

Reviewer #1: All comments have been addressed

Reviewer #2: (No Response)

2. Is the manuscript technically sound, and do the data support the conclusions?

Reviewer #1: Yes

Reviewer #2: Yes

3. Has the statistical analysis been performed appropriately and rigorously?

Reviewer #1: Yes

Reviewer #2: Yes

4. Have the authors made all data underlying the findings in their manuscript fully available?

Reviewer #1: Yes

Reviewer #2: Yes

5. Is the manuscript presented in an intelligible fashion and written in standard English?

Reviewer #1: Yes

Reviewer #2: Yes

Reviewer #1: The manuscript has been revised well but the title looks weird. Could you please change the title something like "Association of Inflammatory Bowel Disease and Diet with Vaginal Immune Responses and Microbiota During Pregnancy" or or something reflecting over all the hypothesis of manuscript? Thank you

Reviewer #2: Manuscript: Association of Inflammatory Bowel Disease and Diet on IL-4, IL-6, IL-8, IL-17, and IFN-γ Vaginal Cytokines and Microbiota During Pregnancy

Round 2 Comments:

Overall comments: The changes have greatly improved the quality of the Manuscript and I would like to thank the authors for incorporating comments. In this draft, I have provided direct suggestions for to expediate the process of re-submission.

In addition to the edits, during review I noticed few minor inaccuracies which were present in draft 1 as well as this draft. Incorporating all the comments will greatly enhance the quality of manuscript.

Specific comments:

Line 1-2: the re-written title is now grammatically incorrect. Association is always “between/with”; it’s not “on”. Please use either or whatever you prefer to give a clearer meaning: Association of Inflammatory Bowel Disease and Diet with IL-4, IL-6, IL-8, IL-17, and IFN-γ Cytokines and Microbiota in Vaginal Mucosa During Pregnancy

OR

Association of vaginal inflammatory markers such as Il-4, IL-6, IL-8, IL-17 and IFN-g and diet with IBD and vaginal microbiota in pregnant individuals

Line 38 – line “Most of the IBD cases were in remission (56.5%)”. It should be A little over half OR mention that 56.5% cases were in remission. 56% would not be considered as “most” .

Line 40 – change cases to individuals.

Line 46 – word respectively can be removed.

Line 47: “Vaginal microbiome was not associated with IBD status in this cohort” - not able to understand this sentence. – which cohort? What part of vaginal microbiome was not associated?

Line 49-50: “Regardless of IBD status, healthier diets are positively associated with an increased abundance of the beneficial L. crispatus in the vagina”. Results section of abstract does not mention any results to support this line (the part which says regardless of IBD status) in conclusion.

Introduction: Line 72: change diet to dietary

Methods: Line 130: “as we have previously done” – do you mean “previously mentioned”?

Line 133: “as we previously described” – grammatically incorrect. In this sentence it should be “as described previously”.

Line 134: “diet quality used to assess 134 how well 13 groups of foods align with key recommendations” – grammatically incorrect. Should be “diet quality used to assess alignment of 13 food groups with key recommendations…”

Line 169-172: grammatical errors in sentence formation. Should be “Microbial taxa (at the species level) and their association with clinical 170 variables were assessed using MaAsLin2’s [40]. Assessment was conducted only for the variables that were found to be significant or marginally significant in the beta diversity analyses and included only the taxa that was present in at least 20% of the samples.”

Results: Table 1: why are p values presented? P values are used to test a hypothesis. In comparing baseline data the idea is that groups are comparable. Do the authors aim to show non-significant differences between the group? In that case you can mention that all groups were comparable and variables did not have significant differences. However, I request authors to justify the reason for using P values before accepting my suggestion.

Line 213: I understand that this study is part of larger MELODY group, and hence individuals in this study are also technically a “cohort of MELODY”. However, for this publication we are talking about 48 individuals and cohorts are IBD and HC – so cohort becomes misleading, it might be better to use the term “study”.

Line 217: grammatical error – “not differences” to be changed to “no differences”

Lines 220-221: MAJOR COMMENT: in previous draft the TNF-a results were not presented and IL-6 results were segregated by Simpson and Shannon analysis. In this draft, TNF-a results are newly introduced. IL-6 results are non-significant for Shannon analysis and significant only for Simpson analysis. Please specify this clearly. The way it is worded is misleading.

Also, I am not sure why the p-values in draft 1 and R1 are different for IL-6, figure 2 (draft 1 0.032 and 0.126 for Shannon and Simpson respectively versus ns and 0.47 for Shannon and Simpson in R1). Please Justify.

Line 221-224: can this be presented after line 218? Suggest to write in same order as presented in figure 1.

Line 236: include TNF-a. i.e. higher IL-6 and TNF-a are associated with….

Line 254 and Line 256: MAJOR COMMENT: Figures show data and p values for Simson and Shannon analysis separately, but in line 254 it is presented as combined? P is ns for Simpson for added sugar intake, and 0.31 for Shannon diversity index. Please specific – current way of writing is misleading.

Line 294: Again, the word study might be more suitable instead of cohort.

Line 239 and 299: MAJOR COMMENT: titles are incorrect. It is vaginal microbiota associated with diet or vaginal cytokines associated with diet. I understand that the authors want to highlight Association of diet with result outcomes – so suggest to use the titles: “Association of diet with……”

Line 308: were negatively associated “with” expression of

Line 309: word respectively can be removed

Discussion: Line 332: replace word most with “more than half or 56.5%”

Enjoyed reading the discussion, especially the limitations paragraph.

**Do you want your identity to be public for this peer review?** For information about this choice, including consent withdrawal, please see our Privacy Policy

Reviewer #1: No

Reviewer #2: **Yes: ** Amruta A Jambekar

---

## [Author Response · Author response to Decision Letter 2]

22 Sep 2025

Please find submitted the complete response to reviewers point by point. Again, thank you for all the feedback.

---

## [Decision Letter · Decision Letter 2]

7 Oct 2025

Association of Vaginal IL-4, IL-6, IL-8, IL-17, IFN-γ, and Dietary Intake with IBD Status and Vaginal Microbiota in Pregnant Individuals

PONE-D-25-04775R2

Dear authors,

I am pleased to inform you that both reviewers enjoyed the manuscript very much and endorsed the revised manuscript for publication.

Thank you for choosing Plos ONE journal to publish your study.

Best regards,

António Machado

Reviewers' comments:

Reviewer's Responses to Questions

**Comments to the Author**

Reviewer #1: All comments have been addressed

Reviewer #2: All comments have been addressed

2. Is the manuscript technically sound, and do the data support the conclusions?

Reviewer #1: Yes

Reviewer #2: Yes

3. Has the statistical analysis been performed appropriately and rigorously?

Reviewer #1: Yes

Reviewer #2: Yes

4. Have the authors made all data underlying the findings in their manuscript fully available?

Reviewer #1: Yes

Reviewer #2: Yes

5. Is the manuscript presented in an intelligible fashion and written in standard English?

Reviewer #1: Yes

Reviewer #2: Yes

Reviewer #1: (No Response)

Reviewer #2: Thank you for providing explanation regarding the calculations of P values for Shannon and Simpson analysis of IL-6 and TNF-a data.

All the best for your publication!

**Do you want your identity to be public for this peer review?** For information about this choice, including consent withdrawal, please see our Privacy Policy

Reviewer #1: No

Reviewer #2: **Yes: ** Amruta Jambekar

---

## [Editor Report · Acceptance letter]

PONE-D-25-04775R2

PLOS ONE

Dear Dr. Maldonado-Contreras,

I'm pleased to inform you that your manuscript has been deemed suitable for publication in PLOS ONE. Congratulations! Your manuscript is now being handed over to our production team.

Kind regards,

on behalf of

Dr. António Machado

Academic Editor

PLOS ONE